

# Impact of stratospheric aerosol intervention geoengineering on surface air temperature in China: A surface energy budget perspective

Zhaochen Liu[1, 4], Xianmei Lang[1, 2, 3], and Dabang Jiang[1, 3, 4*]

5    [1]Institute of Atmospheric Physics, Chinese Academy of Sciences, Beijing 100029, China

[2]Collaborative Innovation Center on Forecast and Evaluation of Meteorological Disasters, Nanjing University of Information Science and Technology, Nanjing 210044, China

[3]CAS Center for Excellence in Tibetan Plateau Earth Sciences, Beijing 100101, China

[4]College of Earth and Planetary Sciences, University of Chinese Academy of Sciences, Beijing 100049, China

10    *Correspondence to*: Dabang Jiang (jiangdb@mail.iap.ac.cn)





**Abstract.** Stratospheric aerosol intervention (SAI) geoengineering is a rapid, effective, and promising means to counteract anthropogenic global warming, but the climate response to SAI, with great regional disparities, remains uncertain. In this study, we use Geoengineering Model Intercomparison Project G4 experiment simulations from three models (HadGEM2-ES, MIROC-ESM, and MIROC-ESM-CHEM) that offset anthropogenic forcing under medium-low emissions (RCP4.5) by injecting a certain amount of $SO_2$ into the stratosphere every year, to investigate the surface air temperature response to SAI geoengineering over China. It has been shown that the SAI leads to surface cooling over China over the last 40 years of injection simulation (2030–2069), which varies among models, regions and seasons. The spatial pattern of SAI-induced temperature changes over China is mainly due to net surface shortwave radiation changes. We find that changes in solar radiation modification strength, surface albedo, atmospheric water vapor and cloudiness affect surface shortwave radiation. In summer, the increased cloud cover in some regions reduces net surface shortwave radiation, causing strong surface cooling. In winter, both the strong cooling in all three models and the abnormal warming in MIROC-ESM are related to surface albedo changes. Our results suggest that cloud and land surface processes in models may dominate the spatial pattern of SAI-induced surface air temperature changes over China.

# 1 Introduction

The increasing anthropogenic greenhouse gas (GHG) concentrations since the industrial revolution have led to global warming. Although the international community has realized the risk of global warming and attempted to reduce GHG emissions, global GHG emissions still show a continuous increase (United Nations Environment Programme, 2020). The "2°C global temperature target" in the Paris Agreements will be unachievable if the current increasing emission trend persists (e.g., Robiou du Pont and Meinshausen 2018). Geoengineering, which aims to counteract global warming by deliberately changing the climate system, is therefore of great research interest. Geoengineering schemes are generally classified into two major types: carbon dioxide removal (CDR) geoengineering by reducing atmospheric carbon dioxide concentration, and solar radiation modification (SRM) geoengineering by increasing planetary albedo. Various specific techniques have been proposed to perform SRM geoengineering, such as injecting sulfate aerosols into the stratosphere (Budyko, 1977), placing shields or deflectors in space (Seifritz, 1989), brightening marine clouds (Latham, 1990), and thinning cirrus clouds (Mitchell and Finnegan, 2009). The method of injecting sulfate aerosols or their precursors into the stratosphere, also known as stratospheric aerosol intervention (SAI) geoengineering, is designed to cool the surface by using these aerosols to reflect and scatter solar radiation (Crutzen, 2006; Wigley, 2006). SAI geoengineering is considered the most promising SRM method due to its high effectiveness, affordability, and timeliness (Shepherd et al., 2009).



SRM geoengineering has not been implemented in reality because of its potential risks and immature technology. The

primary means of recognizing the climate response to geoengineering is by simulating via general circulation models

(GCMs). However, the results from early simulations could not be proved robust due to the differences in experimental

schemes. The Geoengineering Model Intercomparison Project (GeoMIP) has been proposed to address that issue (Kravitz

et al., 2011; 2015). To date, the GeoMIP has designed 12 experiments, including solar dimming, stratospheric aerosol

intervention, marine cloud brightening, and cirrus thinning geoengineering in Coupled Model Intercomparison Project

Phases 5 and 6 (CMIP5 and CMIP6). The GeoMIP provides detailed simulating guidelines for each model and experiment

and calls for all the modeling groups worldwide to become involved and share their simulations. A total of 19 GCMs have

participated in the GeoMIP to date. More detailed information is accessible from the GeoMIP website

(http://climate.envsci.rutgers.edu/GeoMIP/).

Previous studies have indicated that SRM geoengineering could counteract or even reverse anthropogenic global

warming and reduce sea ice melting and thermosteric sea-level rise, as well as the decreasing the frequency and intensity

of extreme temperature and precipitation events (Rasch et al., 2008; Robock et al., 2015; Irvine et al., 2016; Ji et al., 2018;

Jones et al., 2018). It would also come with risks. First, SRM geoengineering reduces the global mean precipitation and

monsoon precipitation and slows the hydrological cycle (Bala et al., 2008; Tilmes et al., 2013; Sun et al., 2020). Second,

SRM geoengineering might induce overcooling of the tropics and undercooling of the poles, which is related to the

difference between the solar and $CO_2$ forcings (Russotto and Ackerman, 2018). Finally, the sudden termination of

geoengineering would lead to a more rapid increase in temperature than the non-geoengineered case (Matthews and

Caldeira, 2007; Jones et al., 2013). Moreover, for SAI geoengineering, the resultant cooling and enhanced polar vortex

might cause stratospheric ozone depletion and thus increase ultraviolet radiation (UV) at the surface (Tilmes et al., 2008;

Eastham et al., 2018).

SRM geoengineering could lead to global cooling, but its regional effects might be different mainly due to the spatially

heterogeneous reduction in solar radiation. This means that if SRM geoengineering was performed, some regions might

face greater climatic impacts or risks than others (Ricke et al., 2013; Kravitz et al., 2014). For example, Robock et al. (2008)

indicated that the SAI-induced weakening of the Asian and African summer monsoons would decrease cloudiness and in

turn warm the surface over northern Africa and India. In addition to the effect of cloudiness, changes in atmospheric

moisture and surface conditions caused by SAI also impact surface air temperature (Kashimura et al., 2017). Large volcanic

eruptions, which inject massive volcanic aerosols into the stratosphere, are considered a natural analog to SAI

geoengineering (Trenberth and Dai, 2007). For instance, the 1815 Mt. Tambora eruption led to the "year without a summer"

over China (e.g., Raible et al., 2016). However, the temperature response to the SAI geoengineering over China has not yet



been studied (Cao et al., 2015).

In this study, we investigate the impact of the SAI geoengineering on the surface air temperature over China and the underlying physical processes from a surface energy perspective. Section 2 provides a brief introduction to the experiments, model data, and decomposition method of net surface shortwave radiation. Section 3 evaluates the ability of models to reproduce the climatological temperature over China in summer and winter. Section 4 presents the summer and winter temperature changes and associated reasons over China in response to SAI geoengineering, and we also analyze the

physical processes responsible for the SAI-induced net surface shortwave radiation changes over China. Conclusions and discussion are presented in Sect. 5.

## 2 Experiments, data, and methods

### 2.1 Experiments

We use the simulations in the G4 experiment from the first phase of the GeoMIP. As a SAI-based geoengineering

experiment, G4 is designed to inject $SO_2$ into the low-level equatorial stratosphere at a consistent rate of 5 Tg per year under the background scenario of Representative Concentration Pathway 4.5 (RCP4.5) (Taylor et al., 2012). This injection rate is equivalent to a case in which the 1991 Mt. Pinatubo eruption occurred every four years (Bluth et al., 1992). The injection period is from 2020 to 2069, and then the experiment continues to run until 2089 to examine the termination effect (Jones et al., 2013). The RCP4.5 simulation for the same period is used as a baseline (non-geoengineered) state. In addition,

the historical simulation for 1986–2005 is applied to evaluate the ability of the selected models to reproduce the climatology of surface air temperature over China.

### 2.2 Data

A total of nine GCMs participated in the G4 experiment, and four of them are available on the Earth System Grid Federation (ESGF), including CSIRO-Mk3L-1-2, HadGEM2-ES, MIROC-ESM, and MIROC-ESM-CHEM (Kravitz et al.,

2013a). Note that CSIRO-Mk3L-1-2 is not selected because the clear-sky shortwave radiation flux at the surface is not available in its outputs (Phipps et al., 2011). Simulations from the other three models are applied for analyses. Monthly datasets are used and calculated as the averages in summer (June–July–August, JJA) and winter (December–January– February, DJF). Considering the intermodel scatter in the temperature response to SAI in the G4 experiment, we analyze the results of each model separately (Yu et al., 2015; Ji et al., 2018). The CN05.1 observation dataset (Wu and Gao, 2013)



is used to evaluate the ability of models to reproduce the climatology of temperature over China. All the observations and

model outputs are interpolated to a common grid with a mid-range horizontal resolution (2.5° longitude by 2° latitude).

A brief description of the models used is illustrated in Table 1. In addition to differences in the physical and chemical

modules related to sulfate aerosol particles, the models have different $SO_2$ injection treatments. For HadGEM2-ES, the

CLASSIC aerosol module (Bellouin et al., 2011) used in the stratosphere makes it possible to handle the injections of $SO_2$,

allowing HadGEM2-ES to finish a complete simulation including the generation and transportation of stratospheric sulfate

aerosols. The injection point is located on the equator (0° longitude), and the injection altitude ranges from 16 to 25 km.

For MIROC-ESM and MIROC-ESM-CHEM, the SPRINTARS aerosol module mainly focuses on tropospheric aerosols.

The prescribed distribution of stratospheric sulfate aerosol optical depth (AOD), according to Sato (2006), is used to drive

the G4 experiment. The only difference between MIROC-ESM and MIROC-ESM-CHEM is that the latter is coupled with

the CHASER atmospheric chemistry module, which can be used to calculate the surface density of sulfate aerosols (Sudo

et al., 2002; Kravitz et al., 2013a).

### 2.3 Decomposition method for SAI-induced shortwave radiation changes at the surface

The surface air temperature change depends on the components of the surface energy budget, including shortwave

and longwave radiation (SW and LW) and sensible and latent heat (SH and LH) (Boer, 1993). For example, the surface

radiation (SW and LW) changes due to SRM geoengineering may be balanced by the surface temperature and/or

nonradiative (SH and LH) flux changes (Andrew et al., 2009). Previous studies indicated that SRM geoengineering could

reduce the surface SW, which was mainly compensated by the decreased LH flux (e.g., Schmidt et al., 2012). Therefore, it

is important to analyze the surface SW response to SAI forcing in this study.

A decomposition method for the SAI-induced net surface SW changes proposed by Kashimura et al. (2017) is applied

for this study. That method is based on the single-layer atmospheric model of SW transfer according to Donohoe and

Battisti (2011) which assumes that the transportation processes of SW, including atmospheric reflection, atmospheric

absorption, and surface reflection, are isotropic. As detailed by Kashimura et al. (2017), the upward SW at the top of the

atmosphere (TOA) and the upward and downward SW at the surface at each grid point can be approximated as:

$$\mathrm{SW}_{\mathrm{TOA}}^{\uparrow} = SR + S\alpha \frac{(1-R-A)^2}{1-\alpha R} \tag{1}$$

$$\mathrm{SW}_{\mathrm{SURF}}^{\downarrow} = S \frac{(1-R-A)}{1-\alpha R} \tag{2}$$





$$SW^{\uparrow}_{SURF} = \alpha SW^{\downarrow}_{SURF} = \alpha S \frac{(1-R-A)}{1-\alpha R} \tag{3}$$

where $S$ is the downward SW at the TOA ($SW^{\downarrow}_{TOA}$), $R$ is the fraction of reflection, $A$ is the fraction of absorption during

SW passing through the atmosphere, and $\alpha$ is the surface albedo. Considering that the four components of SW flux above

($S$, $SW^{\uparrow}_{TOA}$, $SW^{\downarrow}_{SURF}$ and $SW^{\uparrow}_{SURF}$) are directly available from model outputs, $R$, $A$, and $\alpha$ can be calculated from Eqs.

(1)–(3) as follows:

$$R = \frac{S \cdot SW^{\uparrow}_{TOA} - SW^{\downarrow}_{SURF} SW^{\uparrow}_{SURF}}{S^2 - SW^{\uparrow 2}_{SURF}} \tag{4}$$

$$A = (1-R) - \frac{SW^{\downarrow}_{SURF}}{S}(1-\alpha R) \tag{5}$$

$$\alpha = \frac{SW^{\uparrow}_{SURF}}{SW^{\downarrow}_{SURF}} \tag{6}$$

It is noticeable that both $R$ and $A$ are affected by cloud cover. That is, both of them are all-sky values ($R^{as}$ and $A^{as}$).

The clear-sky values of $R$ and $A$ ($R^{cs}$ and $A^{cs}$) are calculated using the clear-sky values of $SW^{\uparrow}_{TOA}$, $SW^{\downarrow}_{SURF}$ and

$SW^{\uparrow}_{SURF}$ to separate the effect of clouds. The effect of clouds, which is denoted as "cl", is calculated from the difference

between all-sky and clear-sky values (e.g., $R^{cl} \equiv R^{as} - R^{cs}$). In addition, we assume that the impact of clouds on surface

albedo ($\alpha$) is negligible. The values of surface albedo are uniformly calculated using the all-sky values of SW collected at

the surface in this study. Here, the flux is defined as downward positive, and the net surface SW at each grid point can be

represented as a function of $S$, $R^{as}$, $A^{as}$, $R^{cs}$, $A^{cs}$, and $\alpha$ as follows:

$$SW^{net}_{SURF} = SW^{\downarrow}_{SURF} - SW^{\uparrow}_{SURF}$$

$$= (1-\alpha)S \left[ \frac{1-(R^{cs}+R^{cl})-(A^{cs}+A^{cl})}{1-\alpha(R^{cs}+R^{cl})} \right] \tag{7}$$

As shown in Fig. 1, the effects of SAI on the net surface SW can be divided into four parts: effects of changes in the

strength of solar radiation modification ($SW_{SRM}$), amount of atmospheric water vapor ($SW_{WV}$), cloud cover ($SW_C$), and

surface albedo ($SW_{SA}$). Here we further assume that the changes in solar radiation modification strength and water vapor

amount would only lead to changes in $R^{cs}$ and $A^{cs}$ respectively, and the concentrations of other atmospheric compositions

related to $R^{cs}$ and $A^{cs}$ would not be affected by SAI. The net surface SW change can therefore be decomposed as follows:

$$SW_{SRM} \equiv SW^{net}_{SURF}(S, R^{cs}_{G4}, R^{cl}_{RCP}, A^{cs}_{RCP}, A^{cl}_{RCP}, \alpha_{RCP}) - SW^{net}_{SURF}(RCP) \tag{8}$$



$$SW_{WV} \equiv SW_{SURF}^{net}(S, R_{RCP}^{cs}, R_{RCP}^{cl}, A_{G4}^{cs}, A_{RCP}^{cl}, \alpha_{RCP}) - SW_{SURF}^{net}(RCP) \tag{9}$$

$$SW_C \equiv SW_{SURF}^{net}(S, R_{RCP}^{cs}, R_{G4}^{cl}, A_{RCP}^{cs}, A_{G4}^{cl}, \alpha_{RCP}) - SW_{SURF}^{net}(RCP) \tag{10}$$

$$SW_{SA} \equiv SW_{SURF}^{net}(S, R_{RCP}^{cs}, R_{RCP}^{cl}, A_{RCP}^{cs}, A_{RCP}^{cl}, \alpha_{G4}) - SW_{SURF}^{net}(RCP) \tag{11}$$

Although the net surface SW differences between G4 and RCP4.5 are not precisely equal to the sum of $SW_{SRM}$, $SW_{WV}$, $SW_C$, and $SW_{SA}$ changes due to the assumption of a single-layer model and the nonlinearity of Eq. (7), this method is effective when analyzing the net surface SW change in response to SAI geoengineering.

### 3 Evaluation of the models


The ability of the models to reproduce the surface air temperature over China is evaluated first. As shown in Fig. 2, the spatial correlation coefficient (SCC), standard deviation (SD), and centered root-mean-square error (CRMSE) between the observation and the historical simulation for the climatological temperature over China during 1986–2005 are calculated and illustrated in a Taylor diagram (Taylor, 2001). The SCCs of the models range from 0.88 to 0.95 in summer and from 0.91 to 0.96 in winter. All the SCCs are significant at the 99% level, meaning that the simulated temperature is in good agreement with the observed temperature. The normalized SDs range from 0.95 to 1.07 in summer and from 1.07 to 1.16 in winter. This result indicates that all three models overestimate the spatial variability of temperature except for HadGEM2-ES in summer. The CRMSEs are 0.31–0.51 for summer and 0.33–0.46 for winter. Taken together, the simulations of summer and winter temperatures by selected models are reliable over China. HadGEM2-ES performs better than MIROC-based models, which may be related to its finer horizontal resolution (Jiang et al., 2016).


The spatial distributions of temperature biases over China between simulations and observations are shown in Fig. 3. Compared to the observations, temperature is generally overestimated in summer but underestimated in winter over China according to the regionally averaged values. In MIROC-ESM and MIROC-ESM-CHEM, cold biases mainly occur over the Tibetan Plateau, and warm biases mainly occur in Xinjiang Province in both summer and winter. The spatial variations of the temperature biases in HadGEM2-ES are relatively small relative to those in MIROC-based models. In summer, the temperature is overestimated over northeastern China, northern Xinjiang, and the Tibetan Plateau, but underestimated over areas south of the Yangtze River in HadGEM2-ES (Fig. 3a). In winter, the simulated temperature in HadGEM2-ES is generally lower than that observed over China, with a regionally averaged bias of –3.04°C (Fig. 3d).


### 4 Results



### 4.1 Changes in surface air temperature over China

Figure 4 shows the temporal evolutions of surface air temperature changes in the G4 experiment and RCP4.5 scenario relative to the present climatology (1986–2005) over China. Both the summer and winter temperatures in G4 increase over time, although they are colder than those in RCP4.5. Positive values occur throughout the whole G4 simulation period, excluding the first several years in HadGEM2-ES and MIROC-ESM. The cooling effect of injection of 5 Tg $SO_2$ per year cannot return the climatological temperature over China under RCP4.5 to the present level, but can delay warming for several years. Considering that the feedback response timescale of diffusive ocean heat uptake in climate models is approximately ten years (Jarvis, 2011), simulations representing the last 40 years of injection (2030–2069) are used to examine the temperature response to SAI over China, as done by Kravitz et al. (2013b) and Tilmes et al. (2013). During this period, the warming trends over all of China in the G4 experiment are 0.36–0.42°C decade$^{-1}$ in summer and 0.35–0.48°C decade$^{-1}$ in winter. As shown in Fig. 4, the warming trend differences between G4 and RCP4.5 are small. This indicates that SAI in the G4 experiment has little impact on the warming trend over China caused by GHG concentrations when the climate system reaches a relatively stable state. The changes in temperature over China in G4 compared to RCP4.5 show that the strongest SAI-induced cooling occurs in HadGEM2-ES, with regional averages of –0.96°C in summer and –1.52°C in winter. The SAI-induced cooling in winter is also stronger than that in summer in MIROC-ESM-CHEM, with magnitudes of –0.61°C and –0.42°C, respectively. The cooling effect in winter (–0.56°C) is slightly weaker than that in summer (–0.60°C) in MIROC-ESM.

The spatial patterns of temperature differences between G4 and RCP4.5 over China are illustrated in Fig. 5. In summer, SAI-induced temperature changes are negative everywhere over China in all three models (Fig. 5a–c). In HadGEM2-ES, strong cooling occurs in the Yangtze-Huaihe River Basin and northern Xinjiang with magnitudes of –1.8 to –1.6°C (Fig. 5a). In MIROC-ESM and MIROC-ESM-CHEM, strong cooling with magnitudes of –1.1 to –0.8°C occurs in southeastern China and central Inner Mongolia, and the Tibetan Plateau, respectively (Fig. 5b–c). In winter, the cooling effect of SAI also occurs over all of China in HadGEM2-ES and MIROC-ESM-CHEM, with strong cooling over the Tibetan Plateau and northeastern China (up to –1.7°C), and northern China (up to –0.8°C), respectively (Fig. 5d–e). For MIROC-ESM, although SAI induces significant cooling over the southern Tibetan Plateau and northeastern China, its impact on winter temperature is weaker than –0.1°C and statistically insignificant in most of Central China. Moreover, the SAI induces slight warming over the source region of the Yellow River and the Sichuan Basin (Fig. 5f). The physical processes responsible for SAI-induced cooling or warming will be discussed in the subsequent sections.

### 4.2 Changes in surface energy components over China



We calculate the regional mean changes in surface air temperature, surface radiation, and surface turbulent heat fluxes

due to SAI forcing over China (Fig. 6). A reduced net surface SW and an increased downward surface LH flux occur in all

three models in both summer and winter, although their magnitudes vary. This indicates that the SAI-induced net surface

SW deficit leads to surface cooling, while the deficient surface radiation is partly compensated by the increased downward

LH flux over China. The decrease in clear-sky net surface SW (SW_CS), which is primarily related to the solar radiation

scattering effect by stratospheric sulfate aerosol particles, is the main cause of the decreased SW. In contrast, the changes

in the SW cloud radiative effect (SW_CRE) are positive and relatively small. Similarly, the decreased surface net LW

contributes to surface cooling, excluding that occurred in HadGEM2-ES in winter. The regionally averaged changes in

LW_CS and LW_CRE exhibit a distinctive difference between summer and winter. The decreased net surface LW is caused

by the negative changes in LW_CS in summer, but by the negative changes in LW_CRE in winter.

The spatial patterns of SAI-induced changes in key energy-related variables over China are illustrated in Fig. 7 and

S1–S2. In summer, the changes in net surface SW over China are closely related to those in SW_CRE. The regionally

averaged changes in the cloud cover fraction over China show a consistent increase in all three models, with magnitudes

of 0.10% in HadGEM2-ES, 0.04% in MIROC-ESM, and 0.06% in MIROC-ESM-CHEM. The increase in cloud cover,

which mainly occurs in the Yangtze-Huaihe River Basin and northern Xinjiang in HadGEM2-ES and southeastern China

and central Inner Mongolia in MIROC-ESM, induces a decrease in SW_CRE, leading to a strong cooling over these regions.

Additionally, the deficit of the downward LH flux, especially over the Yangtze-Huaihe River Basin in HadGEM2-ES and

central Inner Mongolia in MIROC-ESM, can increase cloud cover and amplify cooling although the regional mean changes

in LH are positive. For the MIROC-based models, the large positive values of SW_CRE changes over the Tibetan Plateau

are counteracted by decreased SW_CS, inducing a strong cooling in MIROC-ESM-CHEM (Fig. S1b and S2b). In addition,

the spatial patterns of the net surface LW differences over China between G4 and RCP4.5 are consistent with those of the

net surface SW differences in both summer and winter, but with opposite signs. The changes in LW are mainly dominated

by those in LW_CS. One possible explanation is that the strong surface cooling caused by decreased net surface SW

increases upward surface LW, which is enough to counteract the SAI-induced deficit of downward surface LW, leading to

the positive net surface LW change.

In winter, a uniform SAI-induced reduction in cloud cover is found over China, with regional averages of –0.64% in

HadGEM2-ES, –0.69% in MIROC-ESM, and –0.38% in MIROC-ESM-CHEM. This reduction leads to a general increase

in the SW_CRE over China, causing the positive SW change south of the Yangtze River (Fig. 7k and S1k–S2k). In other

areas of China, however, the changes in the net surface SW are closely related to SW_CS. The SAI-induced decrease in

SW_CS leads to strong cooling over the Tibetan Plateau and northeastern China in HadGEM2-ES, northeastern China in





MIROC-ESM, and northern China in MIROC-ESM-CHEM. Moreover, an increase in SW_CS is found over the source

region of the Yellow River and the Sichuan Basin in MIROC-ESM with magnitudes greater than 6 W m$^{-2}$, leading to the

abnormal winter warming mentioned above. Altogether, the spatial pattern of SAI-induced temperature changes over China

is mainly due to those in net surface SW. The deficit of net surface SW which leads to a strong surface cooling is mainly

induced by the decreased SW_CRE in summer and the decreased SW_CS in winter. The exception is the strong surface

cooling in summer over the Tibetan Plateau in MIROC-ESM-CHEM, which is related to the decreased SW_CS. In winter,

abnormal warming is associated with a large positive SW_CS change in response to the SAI forcing.

**4.3 Physical processes responsible for the SAI-induced SW change.**

Kashimura et al. (2017) pointed out that the net surface SW changes can at least explain a cooling of –1.1 to –0.2°C

in response to the SAI forcing in the G4 experiment on a global scale. According to the above results, the spatial patterns

of temperature differences over China between G4 and RCP4.5 are mainly determined by the net surface SW changes. The

cloud radiative changes occurring over China have been discussed above. In this section, we further address other potential

reasons for net surface SW changes by using the aforementioned decomposition method. Spatial patterns and regionally

averaged values of the decomposition results over China are illustrated in Fig. 8 and S3–S4. In response to the SAI forcing,

changes in the solar radiation modification strength and surface albedo lead to a decrease in net surface SW (SW$_{SRM}$ and

SW$_{SA}$), while those in atmospheric water vapor and cloudiness lead to an increase in net surface SW (SW$_{WV}$ and SW$_{C}$)

over China in both summer and winter, although their magnitudes vary among models and seasons. Furthermore, the

decrease in SW$_{SRM}$ is the largest contributor to the decreased net surface SW over China in all models, with magnitudes of

–2.01 to –1.21 W m$^{-2}$ in summer and –3.00 to –1.13 W m$^{-2}$ in winter.

The spatial distributions of the SAI-induced SW$_{SRM}$ and SW$_{WV}$ changes show a general decrease and increase across

China, respectively (Fig. 8 and S3–S4). The latitudinal distributions of the calculated (used in HadGEM2-ES) and

prescribed (used in MIROC-based models) stratospheric AOD changes caused by SAI in the G4 experiment indicate a

uniform increase in stratospheric AOD over China (Fig. 9). Note that the stratospheric AOD change in HadGEM2-ES is

unavailable, and the tropospheric and stratospheric AOD change is therefore considered as a reasonable alternative (e.g.,

Bellouin et al., 2011). The increased AOD reflects the increased amount of stratospheric sulfate aerosol particles which

leads to the decrease in the SW$_{SRM}$. The results also show that the calculated value of the stratospheric AOD change in

HadGEM2-ES is higher than the prescribed value in the MIROC-based models, which may be a primary cause of the

strongest SAI-induced cooling over China in HadGEM2-ES in this study. Conversely, deficits in column-integrated water

vapor over China caused by SAI occur in all models, with magnitudes of –2.28 to –0.88 kg m$^{-2}$ in summer and –0.62 to –



0.27 kg m$^{-2}$ in winter. This reduction contributes to a decrease in atmospheric absorption of solar radiation, leading to the increase in SW$_{WV}$ (Fig. 10a–10f).

In addition to the SW$_{SRM}$ and SW$_{WV}$, SW$_C$ and SW$_{SA}$ also play important roles in surface SW changes. The results indicate that changes in SW$_C$ and SW$_{SA}$ mainly determine the spatial pattern of net surface SW changes caused by SAI over China. The SW$_C$ changes, which are the same as the changes in SW_CRE, are discussed in Sect. 4.2. The SW$_{SA}$ changes are closely related to land surface conditions. The SAI-induced increase in surface albedo in G4 leads to negative SW$_{SA}$ change over China (Fig. 10g–10l). In summer, the values of increased surface albedo are relatively small, with

regional averages of 0.001 in HadGEM2-ES and 0.002 in MIROC-based models. A significantly increased surface albedo of up to 0.026 occurs in the Tibetan Plateau in MIROC-based models, which leads to decreased surface SW_CS and contributes to surface cooling. The regionally averaged increases in surface albedo range from 0.005 to 0.014 in winter. In addition, the aforementioned abnormal warming seen over the source region of the Yellow River and the Sichuan Basin in MIROC-ESM is also closely related to the decreased surface albedo (Fig. 10k). Considering surface albedo can be

reasonably described as a linear function of snow cover fraction (e.g., Qu and Hall, 2007; Li et al., 2016), we further investigate the spatial pattern of differences in snow cover fraction in MIROC-ESM, and find that matches with surface albedo changes over China (Fig. S5; note that model data are not available for the other two models). It suggests that the SAI-induced surface albedo increase due to enlarged snow cover fraction gives rise to net surface SW decrease over China, which in turn has a positive feedback on surface cooling.

**5 Conclusions and discussion**

We analyze the surface air temperature response to SAI forcing over China based on the simulations from the G4 experiment and RCP4.5 scenario in three models (HadGEM2-ES, MIROC-ESM and MIROC-ESM-CHEM). We also discuss the physical processes involved in the temperature response from a surface energy budget perspective. The main conclusions are summarized as follows.

(1) The three models can well reproduce the present climatological surface air temperature over China in both summer and winter. The cooling effect caused by SAI in the G4 experiment cannot return the climatological temperature over China under RCP4.5 to the present level but can delay warming for several years. Although the SAI-induced temperature differences between the G4 and RCP4.5 simulations are negative over China during the simulation period of 2030–2069, the cooling effect varies among models, regions and seasons. SAI leads to a national-scale cooling over China in all models.

(2) The regionally averaged surface radiation changes over China indicate that both the SAI-induced decreases in net



surface SW and LW, except for the increased LW in winter in HadGEM2-ES, contribute to the surface cooling in all three models. In response to the SAI forcing, the spatial patterns of temperature changes over China are mainly induced by SW changes. In summer, the strong cooling in HadGEM2-ES and MIROC-ESM is mainly due to the decreased SW_CRE caused by the cloud cover decrease. The strong surface cooling over the Tibetan Plateau in MIROC-ESM-CHEM is related

to the decreased SW_CS. In winter, the strong cooling in all three models, together with the abnormal warming in MIROC-ESM, is related to changes in SW_CS.

(3) The net surface SW decomposition shows that the increased $SW_{SRM}$ and $SW_{SA}$ and the decreased $SW_{WV}$ and $SW_C$ have positive and negative contributions to the decrease in net surface SW over China, respectively. Generally, $SW_{SRM}$ decreases and $SW_{WV}$ increases in both summer and winter, which are related to the increased stratospheric AOD and

decreased column-integrated water vapor, respectively. The $SW_C$ and $SW_{SA}$ changes mainly determine the spatial patterns of SW changes due to SAI forcing. Moreover, both the strong summer cooling over the Tibetan Plateau in MIROC-ESM-CHEM and the abnormal winter warming in MIROC-ESM are related to the surface albedo changes. The results above are summarized schematically in Fig. 11.

Equatorial stratospheric sulfate aerosol geoengineering can induce global cooling through the transport of Brewer-

Dobson circulation, and also leads to regional inequities in the temperature response due to the complicated processes of aerosol microphysics and stratospheric transport (Kravitz et al., 2019). This means that some areas will face more severe climatic disasters if this kind of geoengineering is implemented. To solve this issue, certain SAI experiments based on the regional injection method are proposed, such as the stratospheric aerosol geoengineering large ensemble project (GLENS) using CESM1(WACCM) (Tilmes et al., 2018). In addition, the uncertainty of the regional climate response to SAI is closely

related to the reliability of the models (Irvine et al., 2016). It is indicated that the CMIP6 GCMs perform better in simulating the temperature over China than CMIP5 GCMs (Jiang et al., 2020). Therefore, the climate response to SAI geoengineering over China based on state-of-the-art GCM experiments merits further study.

*Code and data availability*. The dataset used in this study can be accessed with the following links: https://esgf-node.llnl.gov/search/cmip5/.

*Author contributions*. Dabang Jiang and Zhaochen Liu designed and performed the research. Zhaochen Liu and Xianmei Lang analysed the data. Zhaochen Liu and Dabang Jiang wrote the manuscript. All authors contributed to this study.





*Competing interests.* The authors declare no competing interests.

*Acknowledgments.* We acknowledge the Geoengineering Model Intercomparison Project Steering Committee and the World Climate Research Program's Working Group on Coupled Modelling. We also thank the climate modelling groups

for producing their model outputs. For the MIROC-based models, we thank Toshihiro Nemoto for his help in downloading the GeoMIP outputs. This work was supported by the National Natural Science Foundation of China (41991284) and the Second Tibetan Plateau Scientific Expedition and Research Program (2019QZKK0101).

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





**Table 1**. Main features and references of three models used in this study.

| Model description | HadGEM2-ES | MIROC-ESM | MIROC-ESM-CHEM |
|---|---|---|---|
| Atmospheric resolution (longitude, latitude, and vertical levels) | 1.875° × 1.25°, L38 | ~2.8° × ~2.8°, L80 | ~2.8° × ~2.8°, L80 |
| Experiment and ensemble numbers | historical: 4<br>RCP4.5: 3<br>G4: 3 | historical: 3<br>RCP4.5: 1<br>G4: 1 | historical: 1<br>RCP4.5: 9<br>G4: 9 |
| Stratospheric aerosols | Generated from $SO_2$ injection | Prescribed | Prescribed |
| Aerosol scheme | CLASSIC | SPRINTARS | SPRINTARS |
| Chemistry | UKCA tropospheric chemistry | None | CHASER |
| Reference | Collins et al. 2011 | Watanabe et al. 2011 | Watanabe et al. 2011 |

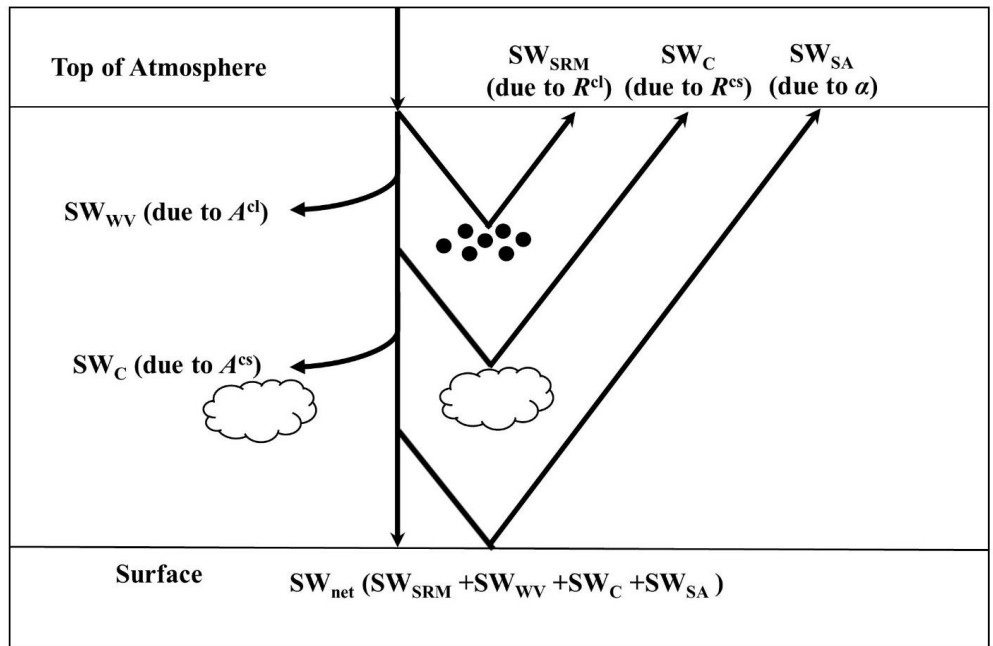

**Figure 1**. Schematic illustration representing the impacts of SAI geoengineering on net shortwave radiation flux at the surface. The $SW_{SRM}$, $SW_{WV}$, $SW_C$, and $SW_{SA}$ represent the changes in shortwave radiation at the surface caused by those in solar radiation modification strength, amount of atmospheric water vapor, cloudiness, and surface albedo, respectively.




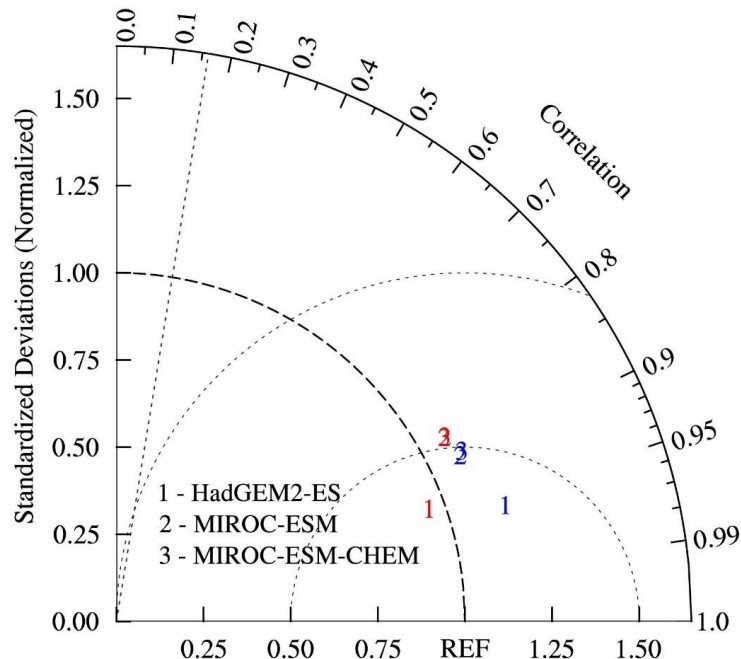

**Figure 2**. Taylor diagram of climatological seasonal temperatures over China between the historical simulations in selected

models and observation during the present period of 1986–2005. Numbers represent individual models. Red and blue

represent summer and winter, respectively. The oblique dotted line shows the 99% confidence level.

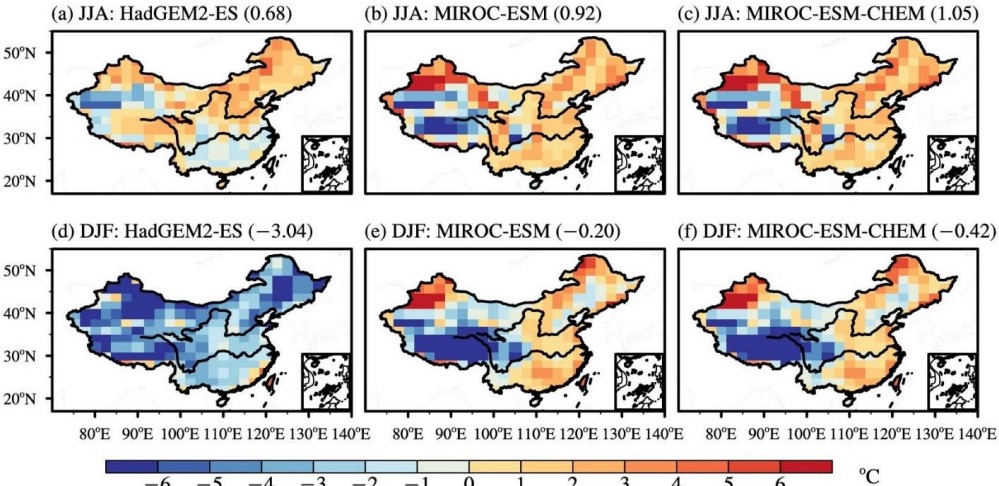

**Figure 3**. Spatial patterns of surface air temperature biases (units: °C) over China between simulations in the historical experiment and observation during the present period of 1986–2005 in (a–c) summer (JJA) and (d–f) winter (DJF). Numbers in parentheses represent regionally averaged values in China.



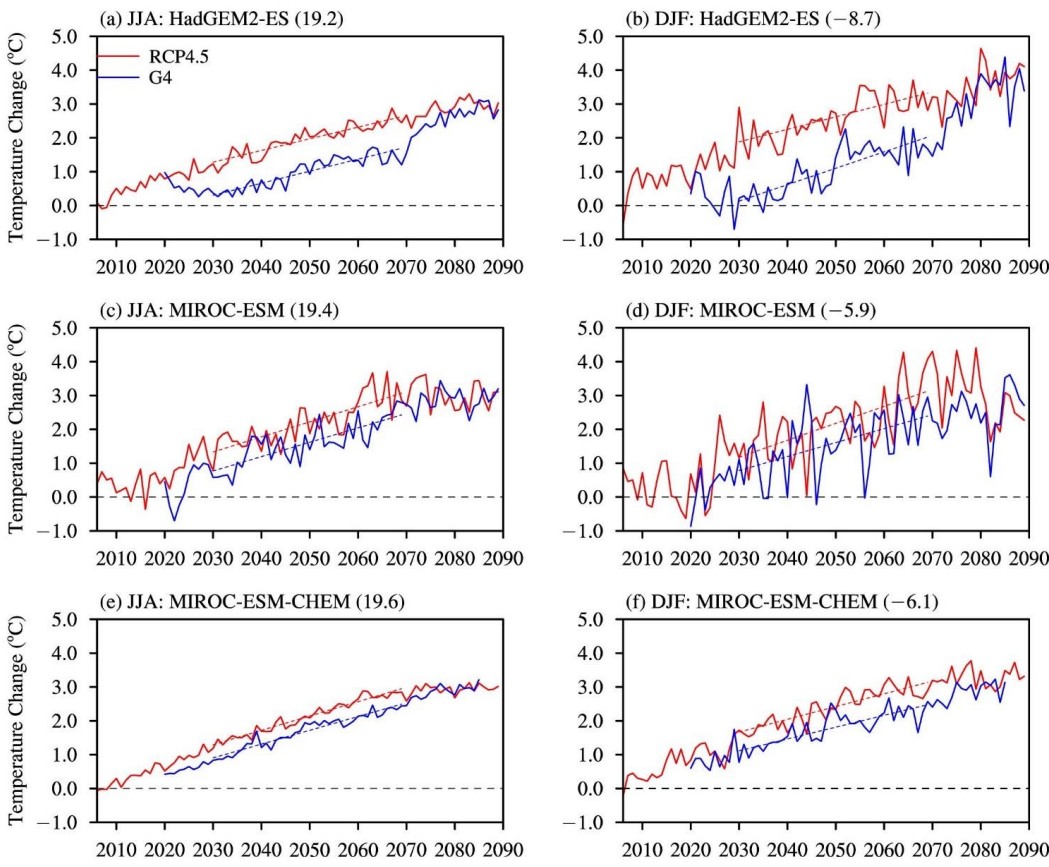


**Figure 4**. Time series of regionally averaged surface air temperature (units: °C) over China in the G4 experiment (solid blue lines) and RCP4.5 scenario (solid red lines) in summer (JJA) and winter (DJF). The values are obtained by subtracting the present climatology (mean of 1986–2005; represented in parentheses) in the historical experiment. Red and blue dashed lines represent the Theil-Sen trends of G4 and RCP4.5 simulations during the period of 2030–2069, respectively.





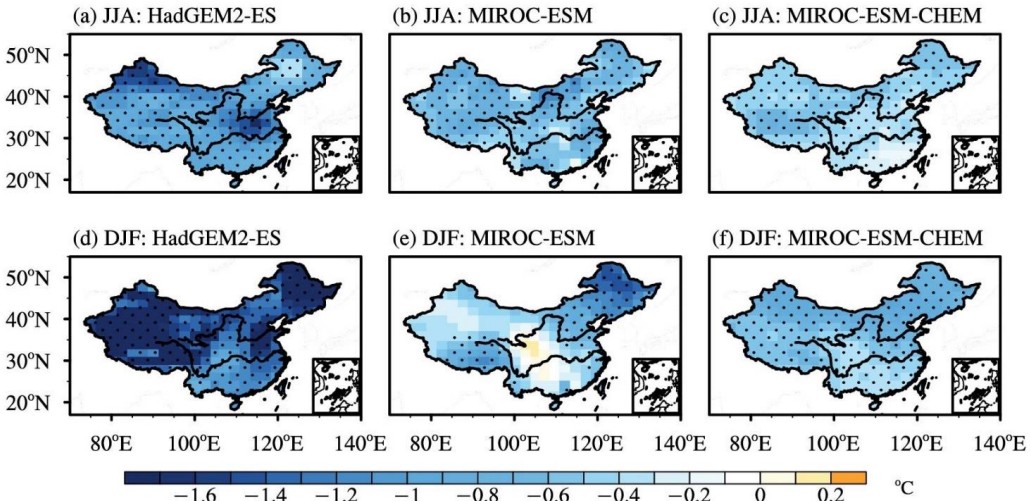


**Figure 5**. Spatial patterns of surface air temperature differences (units: °C) between G4 and RCP4.5 over China during the period of 2030–2069 in (a–c) summer (JJA) and (d–f) winter (DJF). Stippling indicates areas that are statistically significant at the 90% confidence level.

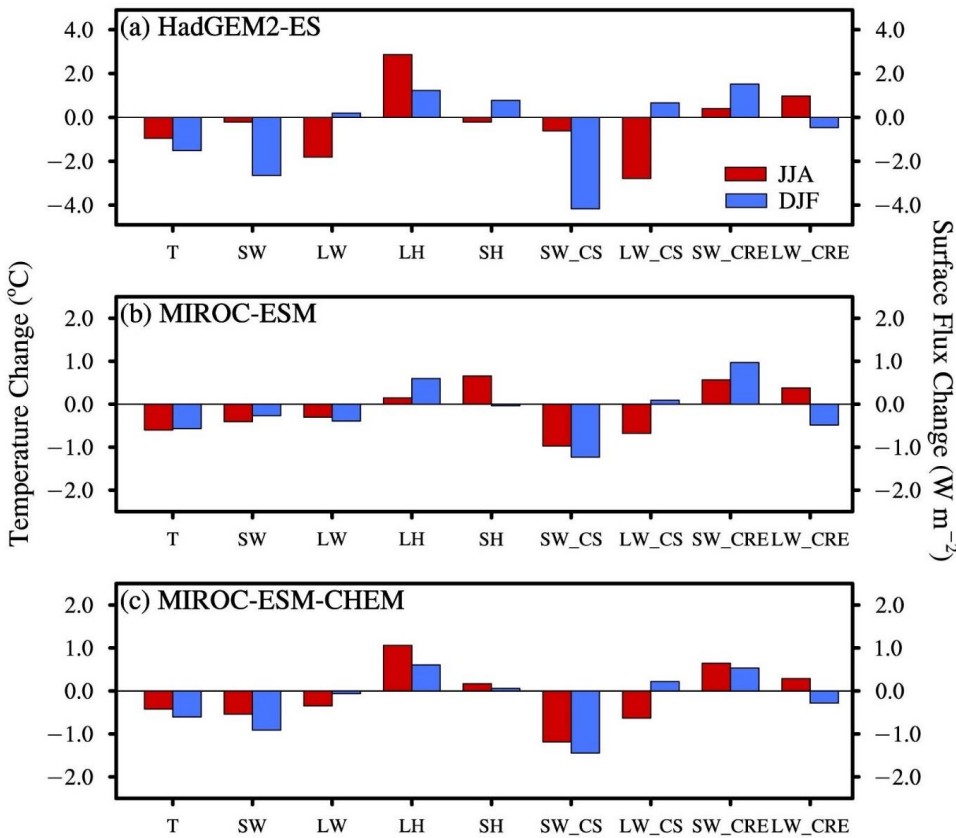

**Figure 6**. Regionally averaged changes in surface air temperature (T, units: °C), net surface shortwave (SW) and longwave (LW) radiation, latent heat (LH) and sensible heat (SH) flux, clear-sky surface shortwave (SW_CS) and longwave (LW_CS) radiation, and cloud radiative effect of shortwave (SW_CRE) and longwave (LW_CRE) in G4 compared to RCP4.5 over China during the period of 2030–2069. Red and blue bars represent values in summer and winter, respectively. Flux is in W m$^{-2}$ with defining as downward positive.

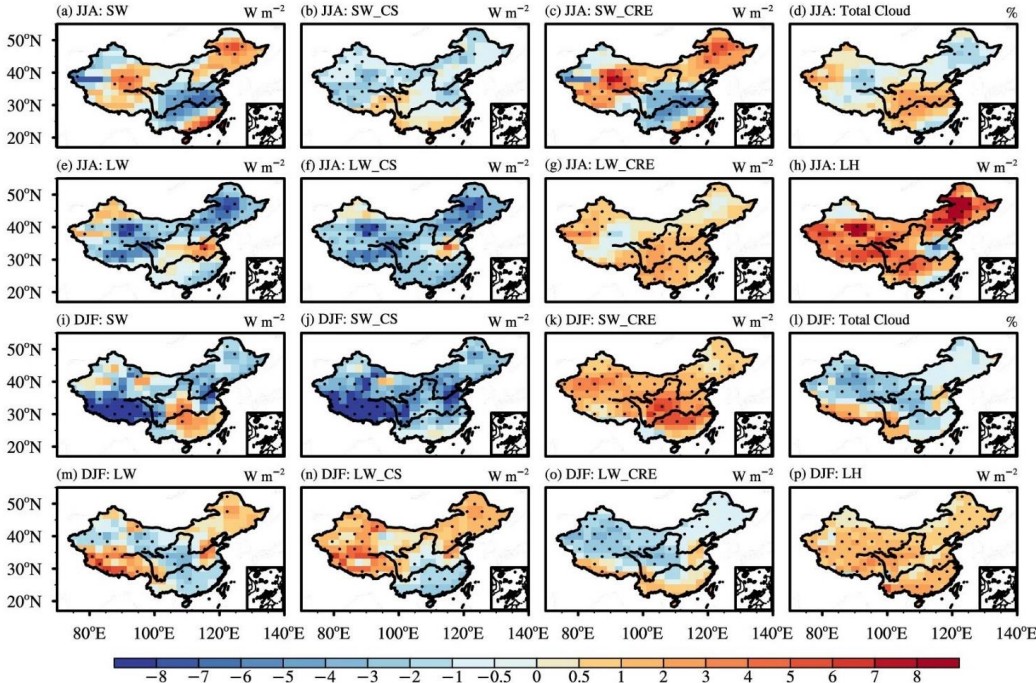


**Figure 7**. Spatial patterns of differences between G4 and RCP4.5 over China for HadGEM2-ES in summer (JJA) and winter (DJF): (a, i) net surface shortwave radiation (SW); (b, j) clear-sky net surface shortwave radiation (SW_CS); (c, k) cloud radiative effect of shortwave (SW_CRE); (d, l) total cloud cover (units: %); (e, m) net surface longwave radiation (LW); (f, n) clear-sky net surface longwave radiation (LW_CS); (g, o) cloud radiative effect of longwave (LW_CRE); (h, p) latent heat flux (LH) during the period of 2030–2069. Flux is in W m$^{-2}$ with defining as downward positive. Stippling indicates areas that are statistically significant at the 90% confidence level.

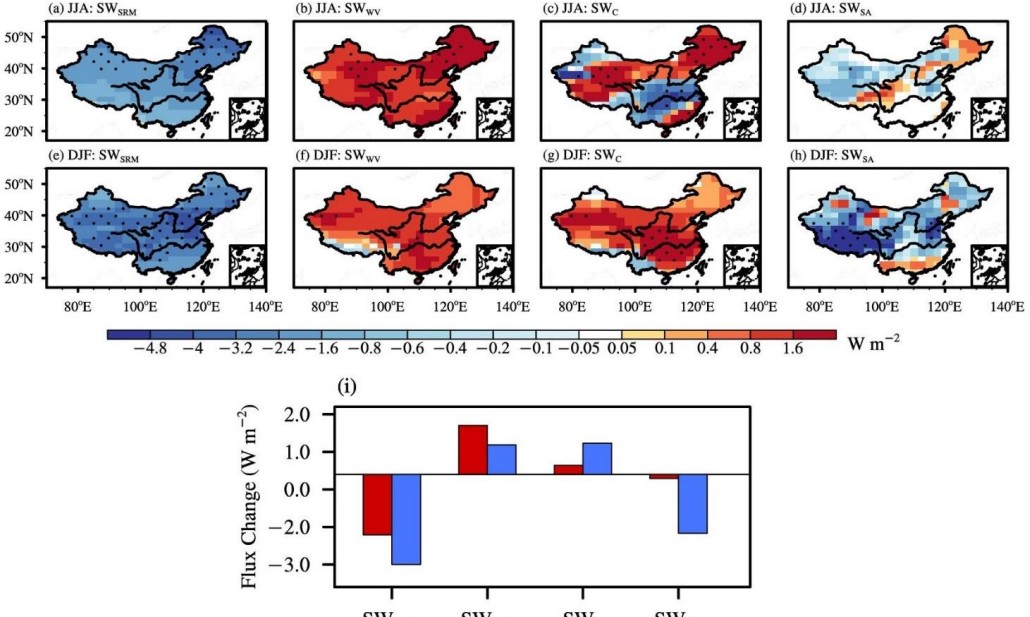

**Figure 8.** Spatial patterns (a–h) and regional mean values (i) of seasonal net surface shortwave radiation changes in G4 compared to RCP4.5 over China for HadGEM2-ES during the period of 2030–2069. The net surface shortwave radiation change is divided into four parts: effect of changes in (a, e) $SW_{SRM}$, (b, f) $SW_{WV}$, (c, g) $SW_C$, and (d, h) $SW_{SA}$. Flux is in W m$^{-2}$ with defining as downward positive. Stippling indicates areas that are statistically significant at the 90% confidence level.

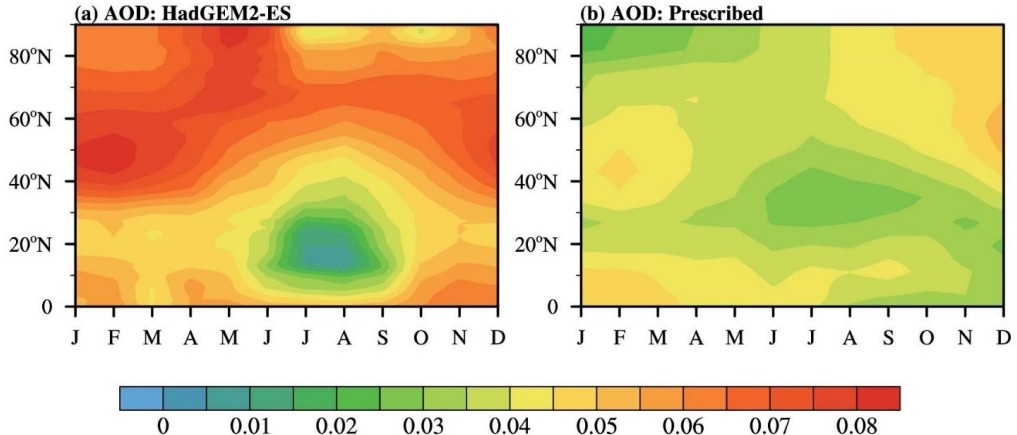

**Figure 9**. Latitudinal distributions of the calculated (a, for HadGEM2-ES) and prescribed (b, for MIROC-ESM and

MIROC-ESM-CHEM) stratospheric AOD changes caused by SAI in G4 experiment over the Northern Hemisphere during

the period of 2030–2069.



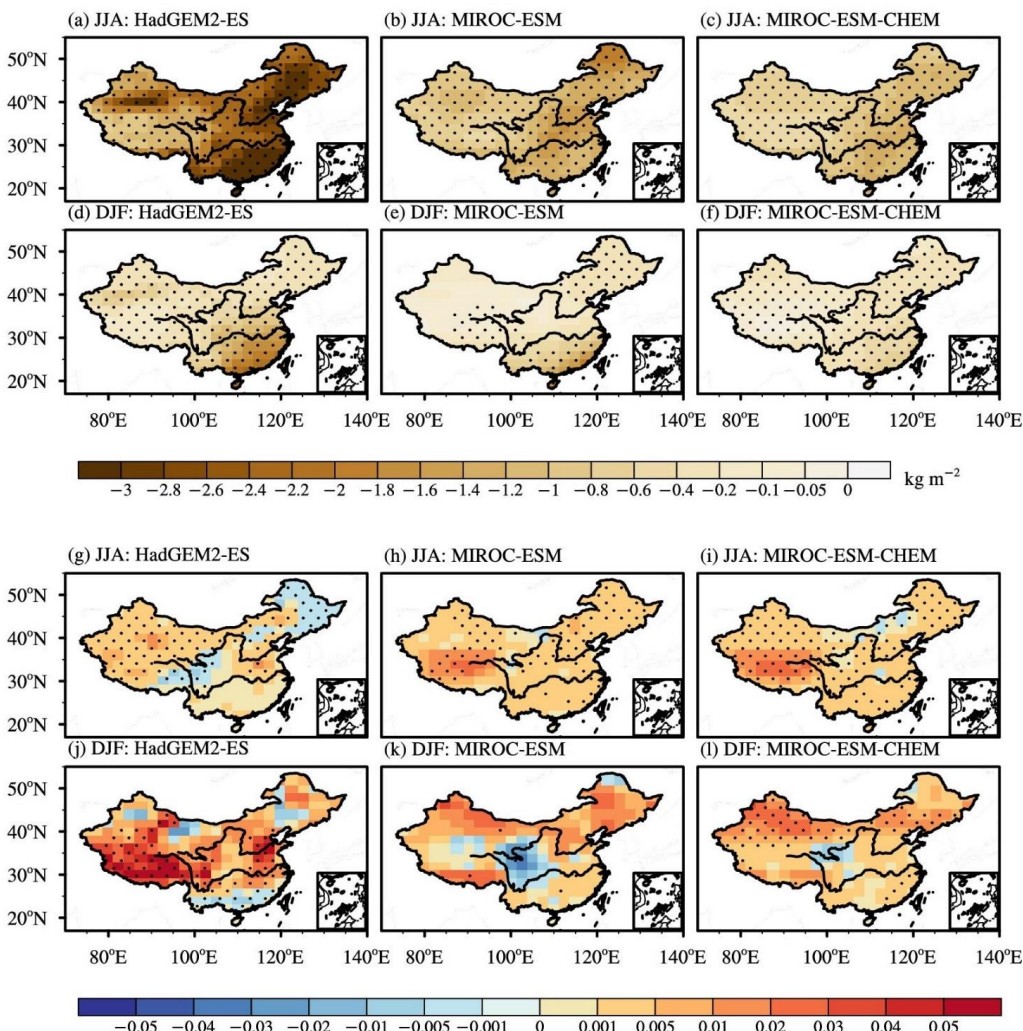

**Figure 10.** Same as Figure 5, but for (a–f) column-integrated water vapor (units: kg m$^{-2}$) and (g–l) surface albedo.





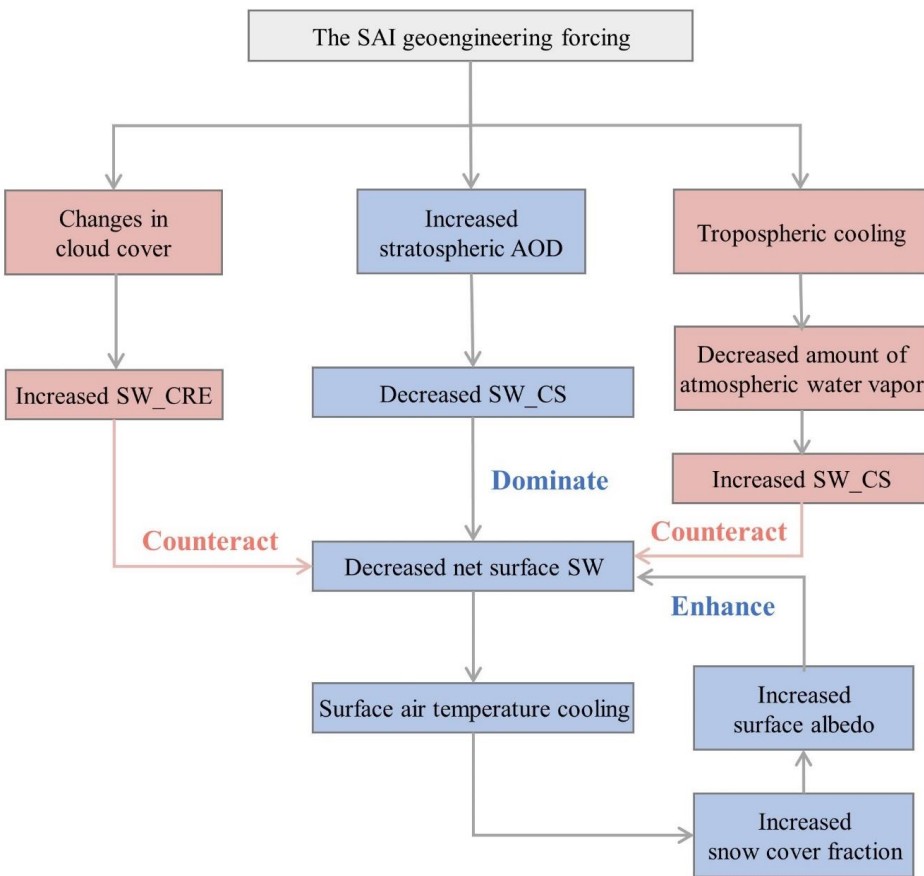

**Figure 11**. Schematic diagram illustrating how the related physical processes impact the surface SW changes over China in response to the SAI forcing in G4 experiment.