# Peer review of "Impact of stratospheric aerosol intervention geoengineering on"

_Atmospheric Chemistry and Physics, 2021_

## Referee Comment (RC3)

[referee-annotated manuscript omitted]

---

## Author Comment (AC1)

**Response to Referee 1**

**Overview:**

This paper analyzed changes in surface air temperature over China in response to stratospheric aerosol injection (SAI) geoengineering. The authors did so by analyzing existing G4 multi-model SAI simulations from the Geoengineering Model Intercomparsion project. The authors used the method of Kashimura et al. (2017) to decompose SAI-induced changes in shortwave radiation into components of solar reduction, albedo change, cloud cover change, and water vapor change. Using this method, the authors further examined spatial pattern of surface shortwave radiation change and its contributors over China.

**Major Comments:**

(1) This paper itself is straightforward and scientifically sound. However, it lacks scientific insight. I see little new scientific findings this paper offers. It essentially applies an existing method and uses existing multi-model dataset to a regional study. To put it in another way, if similar studies are done for another country/region, does it merit another publication?

**Response**: We intend to establish a relationship between changes in surface shortwave radiation and temperature under SAI forcing in the original manuscript. However, as you and Referee 3 suggested, this consideration lacks new scientific findings and has duplication with previous studies. In the revision, we try to diagnose the SAI-induced temperature change over China by using the surface energy balance equation based on Lu and Cai (2009) (Lines 136–166). The method proposed by Kashimura et al. (2017) is only used to diagnose the downward shortwave radiation change under clear sky conditions (Lines 153–166). Results indicate that the SAI-induced temperature change over China is dominated by the robust decreases in downward clear-sky radiation fluxes and associated with the cloud effective forcing and surface albedo feedback changes. The spatial pattern of temperature response over China is mainly related to the shortwave radiative effect of clouds and surface albedo feedback. The physical processes which drive the temperature change have also been discussed in the revision (Lines 265–300). Taken together, we believe that these analyses provide new scientific

findings.

**Specific comments:**

(1) Lines 31-32: Recent reports such as IPCC special reports on 1.5 degree warming treat CDR and SRM separately and no longer lump them together as geoengineering.

**Response**: This sentence has been rewritten as you suggested (Lines 34–36).

(2) Lines 51-52: SAI does not necessarily reduce precipitation. It only reduces global mean precipitation if SAI is used to compensate GHG-induced global mean warming.

**Response**: This sentence has been rewritten in the revision for clarity (Lines 60–62).

(3) Lines 58-59: Possible ozone depletion due to SAI is not a result of SAI induced cooling.

**Response**: The possible ozone depletion is mainly caused by the SAI-induced heterogeneous chemistry change (Tilmes et al., 2008). This sentence has been rewritten for accuracy (Lines 66–68).

(4) Line 66: It is important to note that volcanic eruptions are just imperfect analog of SAI.

**Response**: As you suggested, the volcanic eruption is not a perfect analog of SAI. The sulfate aerosols from massive volcanic eruptions only last for 2–3 years, while the SAI-induced aerosols are continuously replenished for decades or centuries. The relevant explanation has been added to the revision (Lines 82–84).

(5) Lines 70-76: It is not clear what is the novelty of this study. The authors state that no previous studies have analyzed effect of SAI on surface air temperature over China. However, this is a very weak justification of the novelty of this study. What new scientific insight we can obtain from this study? In other words, if someone writes another paper, stating that no

previous study has focused on surface air temperature response over another region/country, can it be justified for publication?

**Response**: Following your comment, firstly, we have emphasized the importance of analyzing the effect of SAI on surface air temperature over China in the revision (Lines 75–79). As an important region in combating climate change, China's attitude to SAI is crucial to the international geoengineering research community. Considering the combined effect of the Tibetan Plateau and the East Asian monsoon, the climate over China would be strongly influenced by SAI geoengineering. It is therefore worth investigating how SAI affects the climate over China.

Secondly, we have changed the research method in the revision. The diagnosis based on surface energy budget quantifies the effect of downward energy fluxes changes on surface air temperature over China under SAI forcing. The related physical processes are also discussed in the revision (Lines 265–300).

(6) Lines 108-113: Everything stated in this paragraph is self-evident.

**Response**: This paragraph has been completely rewritten in the revision for clarity (Lines 129–135).

(7) Lines 114: Is it necessary to repeat the method and equations of Kashimura et al. (2017) in detail? What is new here?

**Response**: As you suggested, the detailed introduction of this method is redundant. This part has been shortened in the revision (Lines 153–166).

(8) Line 222: Why surface cooling increases upward LW radiation?

**Response**: The original "increases" should be "decreases" here. The discussion about upward radiations has been removed in the revision.

**Reference:**

Kashimura, H., Abe, M., Watanabe, S., Sekiya, T., Ji, D., Moore, J. C., Cole, J. N., and Kravitz, B.: Shortwave radiative forcing, rapid adjustment, and feedback to the surface by sulfate geoengineering: Analysis of the Geoengineering Model Intercomparison Project G4 scenario, Atmos. Chem. Phys., 17, 3339–3356, https://doi.org/10.5194/acp-17-3339-2017, 2017.

Lu, J., and Cai, M.: Seasonality of polar surface warming amplification in climate simulations. Geophys. Res. Lett., 36, L16704, https://doi.org/10.1029/2009GL040133, 2009.

Tilmes, S., Müller, R., and Salawitch, R.: The sensitivity of polar ozone depletion to proposed geoengineering schemes, Science, 320, 1201–1204, https://doi.org/10.1126/science.1153966, 2008.

---

## Author Comment (AC2)

**Response to Referee 2**

**Overview:**

This is a carefully done and well written study. I appreciate that the authors are able to use old simulations to do new science. I have a few minor comments and one major one.

**Major Comments:**

(1) My major comment is similar to that of Reviewer #1 about the novelty. I think there are good reasons to look at specific regions and try to understand them better. Such investigations could warrant a new paper if they provide insight. My problem is that the authors have not provided much insight that is specific to China. The analyses they did could easily be applied anywhere in the world. It would be much more useful to add some discussion about something specific to China that requires more in-depth analysis. I won't decide for the authors what they should focus on – there are lots of things to choose from.

**Response**: Following your suggestion, we have emphasized the importance of analyzing the effect of SAI on surface air temperature over China in the revision (Lines 75–79). As the largest developing country in the world, China plays an important role in combating climate change. China's attitude to SAI is crucial to the international geoengineering research community. Considering the combined effect of the Tibetan Plateau and the East Asian monsoon, the climate over China would be strongly influenced by SAI. But few studies have studied the temperature response to SAI geoengineering over China explicitly. In the revision, the revised diagnosis analyses based on surface energy budget quantify the effect of downward energy fluxes changes on surface air temperature over China under SAI forcing.

China is a big country with complex topography. This feature leads to the surface albedo feedback change with large seasonal and spatial variations under SAI forcing. The negative surface albedo feedback related to increased snow cover fraction also amplifies the surface cooling, especially over the Tibetan Plateau in summer, and over northwestern and central China in winter. The land surface processes in models affect the spatial pattern of SAI-induced surface air temperature changes over

China. We believe that this revised analysis provides new scientific insights.

**Specific comments:**

(1) I would appreciate a much more nuanced picture of geoengineering than you're providing. On lines 52-53, you talk about slowing of the hydrologic cycle. That's true, but climate change accelerates the hydrologic cycle, with some pretty bad consequences for a lot of people.

**Response**: We have improved this part in the revision. As you suggested, the SRM geoengineering would reduce the global mean precipitation and monsoon precipitation and slow the hydrological cycle if it is used to offset the GHG-induced global warming. This sentence has been rewritten for clarity (Lines 60–62).

(2) One lines 54-55, you talk about overcooling of the tropics and undercooling of the poles. That is not a foregone conclusion–see Kravitz et al. 2016 (ESD) or 2017 (JGR).

**Response**: This foregone conclusion has been deleted in the revision as you suggested.

(3) On lines 55-57, you talk about termination. That is a risk, but it's less of a risk for lower magnitude deployments of geoengineering and a greater risk for higher magnitude deployments.

**Response**: We agree that the severity of the termination effect depends on the magnitude of geoengineering deployment. This statement has been added to the revision (Lines 65–66).

(4) Your discussion of risks needs appropriate context. Another example is line 62. What you say is true, but it depends on the amount of geoengineering. Irvine et al. (2019) found that with only a little bit of geoengineering, most regions would benefit under a wide variety of metrics.

**Response**: We agree that the appropriate SRM geoengineering may lead to global cooling and benefit most regions. This part has been rewritten accordingly (Lines 69–71).

(5) Line 69: China has been studied in several papers, but not explicitly or in much detail. Also see my major comment above.

**Response**: This sentence has been rewritten for accuracy (Lines 84–85).

(6) Line 175: This is a strawman argument. It wasn't designed to return the temperature to climatological RCP4.5 levels.

**Response**: We intend to indicate that although the injection of 5 Tg $SO_2$ per year can lead to a surface cooling over China, the climatological temperature in G4 is still higher than the present level. This sentence has been rephrased in the revision (Lines 200–201).

(7) Lines 200: Instead of increased downward LH, it should be decreased upward LH. The actual LH doesn't become negative.

**Response**: In the revision, the "downward latent heat flux" statement has been rephrased accordingly (e.g., Line 230).

(8) Lines 204: Why does LW decrease? My guess is water vapor, which you talk about later in the paper. But you should say so here.

**Response**: The decrease in downward surface LW is mainly due to the tropospheric cooling and reduced atmospheric water vapor. The relevant explanation has been added to the revision accordingly (Lines 240–241).

(10) Line 212: These changes are quite small. Or did you mean these to be 10, 4, and 6%, respectively? (Same comment for lines 224-225.) In either case, are the changes statistically

significant?

**Response**: In the original manuscript, the summer cloud cover changes are 0.10%, 0.06% and 0.04% among models exactly. These changes are pretty small and insignificant, which are mainly due to the large spatial dispersion. In the revision, we have mainly discussed the robust change for individual regions (e.g., Lines 243–246). The discussion about the regionally averaged change has been removed.

(11) Line 221: I don't understand this explanation. Cooling should reduce upward LW, not increase it.

**Response**: The original "increases" should be "decreases" here. This sentence has been removed in the revision.

(12) I like Figure 11. I've never seen something that clear before.

**Response**: We are glad you like it. This diagram has been improved in the revision (Fig. 12).

---

## Author Comment (AC3)

**Response to Referee 3**

**Overview:**

I recommend major revisions, due to the reasons below and the 16 comment on the attached annotated manuscript.

**Major Comments:**

(1) Fundamentally, this paper ignores an important component of climate change, and that is advection of energy. Surface temperature does not just depend on vertical energy fluxes, but also on changes of atmospheric circulation. By ignoring impacts on atmospheric circulation and wind patterns, as well as on storms and cloudiness, it ignores fundamental processes of climate change.

**Response**: We agree with your opinion on the fundamental processes of climate change. We would like to explain that the main factor of surface air temperature change under SAI forcing we previously considered is the vertical energy fluxes rather than the advection of energy. Following your comment, in the revision, we have investigated the temperature change by diagnosing the surface energy balance equation (Lines 136–153). Results show that the changes in vertical energy fluxes (including downward surface longwave and shortwave radiations, latent and sensible heat fluxes) dominate the SAI-induced surface cooling over China in both summer and winter (Fig. 8a). In addition, the radiative effect of clouds, which is related to atmospheric circulation, also plays an important role in the temperature change. In summer, the moisture flux convergence increases cloud cover, resulting in a strong local cooling over northwestern and central China (Lines 287–290).

(2) Why does this paper just look at China? With all the data, why don't the authors look at the entire globe?

**Response**: We would like to mention that previous studies have investigated temperature changes from a global-scale perspective (e.g., Niemeier et al., 2013; Kashimura et al., 2017; Ji et al., 2018). In recent

years, increasing attention has been given to the climatic response to solar radiation modification (SRM) on a regional scale, such as over Africa (Pinto et al., 2019; Da-Allada et al., 2020) and America (Xu et al., 2020). As the largest developing country in the world, China plays an important role in combating climate change. China's attitude to geoengineering is crucial to the international geoengineering research community. Considering the combined effect of the Tibetan Plateau and the East Asian monsoon, the climate over China would be strongly influenced by SAI. But the climatic impact has not yet been investigated explicitly so far. It is therefore meaningful to focus on the temperature change over China under SAI forcing. The relevant explanation has been added in the revision (Lines 75–79).

(3) What is the new science? The results are what one would expect. and there is little diagnosis of the reasons for the changes.

**Response**: In the original manuscript, we intend to explain the surface air temperature change by establishing a relationship between changes in surface shortwave radiation and temperature under SAI forcing. In the revision, we have diagnosed the temperature change over China by using the surface energy balance equation (Lines 136–153). The results indicate that the SAI-induced surface cooling over China is dominated by the robust decreases in downward clear-sky radiation fluxes, and associated with the cloud effective forcing and surface albedo feedback changes. The shortwave radiative effect of clouds and the surface albedo feedback determine the spatial pattern of temperature change under SAI forcing. The physical processes which dominate the temperature change have been investigated in Lines 265–300.

(4) The authors only use three models, due to finding the data on ESGF, but the output from the rest of the models could have been obtained from the modeling groups.

**Response**: A total of 12 GCMs participated in the G4 experiment. We would like to explain that six models should not be considered in this study due to their known issues (Lines 106–111). According to your suggestion, we have contacted the modeling groups and obtained the model data. We have added the output from the other three models (BNU-ESM, CanESM2 and CNRM-ESM1) into the revised manuscript as you suggested (Table 1).

(5) I find the algebra and terms in section 2.3 confusing. What is the difference between R and alpha? They are both reflection.

**Response**: Both the $R$ and $\alpha$ are reflections. In this study, the $R$ ($F$ in the revision) represents the fraction of solar radiation reflected by the atmosphere (Line 161). The $\alpha$ is surface albedo. It represents the fraction of solar radiation reflected by the surface (Lines 151–152). We have added the relevant explanations in the revision for clarity.

(6) There is a supplemental file, but it is not referenced at all in the manuscript.

**Response**: We have listed the pertinent results for the MIROC-based models, together with the snow cover fraction change in the original supplement. Those results have been referenced in the original manuscript (Lines 210, 218, 226, 242, 249 and 272). In the revision, the spatial patterns of SAI-induced changes in key energy-related variables over China for the individual models have been illustrated in the supplemental file. Those results have been used to analyze the SAI-induced abnormal warming in the MIROC-based models and referenced in the revision (Lines 248, 259, 287 and 300).

(7) It is great that they evaluate the models first before using them, but although the Taylor diagrams look pretty good, there are still substantial biases.

**Response**: In the revision, the multi-model mean result is better than most models, but the bias still exists. The bias is inherent due to the limited understanding of the real climate system, the non-linear nature of some model equations, and the parameterization for processes. The results in the geographical distribution of simulation (Fig. 3) and the Taylor diagram (Fig. 2) both indicate the selected models can reproduce the climatology of temperature over China. The selected models are therefore reliable in this study.

(8) The manuscript is in quite a small font. In the future, make it larger to make it easier for the reviewers.

**Response**: The font size has been enlarged in the revision as you suggested.

**Specific comments:**

(1) Line 11: The is completely wrong. SAI does not exist, and there is no technology to do it. So "is" is wrong. It is a proposed scheme. Also, how do you know that it would be effective? The technology has never been proven. Is it really possible to produce a cloud of aerosols as modeled? Furthermore, why is it promising? It may produce more risks than it alleviates.

**Response**: We have replaced "a rapid, effective, and promising means" with "a proposed scheme" in the revision as you suggested (Line 12). By the way, the statements of "rapid" and "effective" are based on Table 3.4 in Shepherd et al. (2009), in which they consider that the cooling effect of SAI is feasible and potentially very effective. But it is impossible to simulate the real cloud of aerosols. The statement of "promising" is based on Visioni et al. (2018). They indicate SAI is a promising proposal because of its potential to cool the Earth and its assumed technological feasibility. However, we agree that it is not appropriate to describe SAI qualitatively before this technology is proven as you suggested.

(2) Line 16: "It has been shown" by others previously, or by you? If the latter, change to "We have found".

**Response**: Here we intend to express the result shown by us. We have changed "It has been shown" into "We have found" accordingly (Line 17).

(3) Line 57: No. The main mechanism is heterogeneous chemistry on the injected sulfate aerosols.

**Response**: This sentence has been rephrased in the revision accordingly (Lines 66–68).

(4) Line 70: Why just this region? Why not globally, since you have all the data.

**Response**: We have already answered this question above (please see our reply to Major 2).

(5) Lines 88-89: But the output from the rest can be obtained from the modelers.

**Response**: We have contacted the modeling groups and solved this problem in the revision (please see our reply to Major 4).

(6) Line 90: Delete "Note that".

**Response**: This sentence has been rephrased in the revision (Lines 107–108).

(7) Line 108: But advection is also important. What about changes in atmospheric circulation?

**Response**: We have answered this question above (please see our reply to Major 1).

(8) Lines 164 and 189: Where is the Xinjiang Province? Shown on map. Non-Chinese readers will not be familiar with these. Include on one of your maps the locations of all the Chinese regions you mention.

**Response**: The statements of location have been rephrased so that they can be understood by non-Chinese readers easily. For example, "the source region of the Yellow River and the Sichuan Basin" has been changed into "the upper reaches of the Yellow River and the middle and upper reaches of the Yangtze River" in the revision (Lines 220–221).

(9) Figure 1: Why are there two SW$_C$? You need to define all the terms in the caption. What is A$^{cs}$? What is R$^{cs}$? Why do you multiply SW$_{net}$ by the other terms?

**Response**: The SW$_C$ includes both the effects of changes in SW absorption and reflection rates of cloud. $R^{cs}$ ($F^{cs}$ in the revision) is the fraction of solar radiation reflected by the atmosphere under clear sky conditions, and $A^{cs}$ the fraction of absorption during solar radiation passing through the atmosphere

under clear sky conditions. The relevant definitions have been illustrated in Lines 161–162.

For the net surface SW change ($SW_{net}$), we intend to express that the $SW_{net}$ can be decomposed into four terms. This illustration should be rewritten as "$SW_{net} \approx SW_{SRM} + SW_{WV} + SW_{C} + SW_{SA}$". In the revision, Figure 1 has been removed due to the change in research method.

(10) Line 465: What does "the oblique dotted line" mean? Which line?

**Response**: We refer to the dotted straight line here. This sentence has been rephrased in the revision (Line 539).

(11) Line 474: What is the Theil-Sen trend?

**Response**: The Theil-Sen trend estimation method is a nonparametric technique for estimating the linear trend. In the revision, we have changed the Theil-Sen method into the widely used linear regression method (Figs. 4–5).

(12) Figure 9: Add January on right side of figures, too. So as to plot the entire 12-month seasonal cycle.

**Response**: Figure revised (Fig. 10).

(13) Line 500: "The stratospheric AOD". At what wavelength?

**Response**: The SAOD is determined at 550 nm in this study. This information has been added accordingly (Line 583).

(14) Figure 11: But clouds affect longwave, too.

**Response**: In the original manuscript, we have illustrated how the related physical processes impact

the surface shortwave radiation change in Fig. 11. The schematic diagram has been redrawn to summarize the downward surface radiation changes over China under SAI forcing in the revision. This diagram includes the cloud longwave radiative forcing as you suggested (Fig. 12).

**Reference:**

Da-Allada, C. Y., Baloïtcha, E., Alamou, E. A., Awo, F. M., Bonou, F., Pomalegni, Y., Biao, E. I., Obada, E., Zandagba, S., Tilmes, S., and Irvine, P. J.: Changes in west African summer monsoon precipitation under stratospheric aerosol geoengineering, Earths Future, 8, e2020EF001595, https://doi.org/10.1029/2020EF001595, 2020.

Ji, D., Fang, S., Curry, C., Kashimura, H., Watanabe, S., Cole, J. N., Lenton, A., Muri, H., Kravitz, B., and Moore, J.: Extreme temperature and precipitation response to solar dimming and stratospheric aerosol geoengineering, Atmos. Chem. Phys., 18, 10133–10156, https://doi.org/10.5194/acp-18-10133-2018, 2018.

Kashimura, H., Abe, M., Watanabe, S., Sekiya, T., Ji, D., Moore, J. C., Cole, J. N., and Kravitz, B.: Shortwave radiative forcing, rapid adjustment, and feedback to the surface by sulfate geoengineering: Analysis of the Geoengineering Model Intercomparison Project G4 scenario, Atmos. Chem. Phys., 17, 3339–3356, https://doi.org/10.5194/acp-17-3339-2017, 2017.

Niemeier, U., Schmidt, H., Alterskjaer, K., and Kristjánsson, J. E.: Solar irradiance reduction via climate engineering: Impact of different techniques on the energy balance and the hydrological cycle, J. Geophys. Res.-Atmos., 118, 11905–11917, https://doi.org/10.1002/2013JD020445, 2013.

Pinto, I., Jack, C., Lennard, C., Tilmes, S., and Odoulami, R. C.: Africa's climate response to solar radiation management with stratospheric aerosol, Geophys. Res. Lett., 47, e2019GL086047. https://doi.org/10.1029/2019GL086047, 2020.

Shepherd, J. G.: Geoengineering the Climate: Science, Governance and Uncertainty (Policy Document No. 10/09), London: Royal Society, 82 pp, 2009.

Visioni, D., Pitari, G., di Genova, G., Tilmes, S., and Cionni, I.: Upper tropospheric ice sensitivity to sulfate geoengineering, Atmos. Chem. Phys., 18, 14867–14887, https://doi.org/10.5194/acp-18-14867-2018, 2018.

Xu, Y., Lin, L., Tilmes, S., Dagon, K., Xia, L., Diao, C., Cheng, W., Wang, Z., Simpson, I., and Burnell, L.: Climate engineering to mitigate the projected 21st-century terrestrial drying of the Americas: a direct comparison of carbon capture and sulfur injection, Earth Syst. Dynam., 11, 673–695, https://doi.org/10.5194/esd-11-673-2020, 2020.

---

## Author Comment (AC4)

**Response to Referee 4**

**Overview:**

This paper investigates the effect of the solar radiation modification on the downwelling solar radiation at the surface over China and hence on the surface temperature. For this purpose, it uses the simulation data from the G4 experiment from 3 climate models of the Geoengineering Model Intercomparison Project (GeoMIP). In G4, SO2 is injected into the stratosphere at a rate of 5 Tg SO2 per year on the RCP4.5 emission scenario. The analysis in this paper focuses on the contribution of key processes involved in the reduction of solar radiation at the surface. Four processes are assessed: AOD changes in the atmosphere due SO2 injections, water vapor changes due to tropospheric cooling, changes in clouds and surface albedo. A simple 1-layer atmosphere model is used to facilitate the understanding. This decomposition of changes in downwelling shortwave radiation at the surface by a simple model is an elegant approach that helps to understand the complex interaction between various physical processes in climate models. The presentation is fine but could be improved in the revision. The paper could be accepted for publication after the main and specific comments listed below are addressed.

**Major Comments:**

(1) Clear insight into the contribution of water vapor changes and albedo changes should be provided in the revision. A colder atmosphere holds less water vapor –7% decrease in water vapor per deg C decrease in temperature. Reduced water vapor causes reduced absorption of solar radiation that is coming down. In the case of surface albedo, colder temperatures are likely to lead to an increase in snow on the surface which would reflect more sunlight and hence a reduction in net surface shortwave radiation.

**Response**: In the revision, we have evaluated physical processes responsible for the SAI-induced temperature change based on the surface energy budget equation (Lines 265–300). The decreased tropospheric temperature reduces atmospheric water vapor amount following the Clausius-Clapeyron

relationship as you mentioned. On the one hand, the reduced atmosphere water vapor decreases the downward surface clear-sky LW, contributing to the surface cooling primarily. On the other hand, the reduced water also increases the downward surface clear-sky SW by changing the atmospheric absorption. This effect partly offsets the decreased SW caused by aerosols scattering over China. In addition, the seasonal difference in water vapor change leads to a severer surface cooling in winter over China (Lines 265–282).

Under SAI forcing, the negative surface albedo feedback due to the increased snow cover contributes to the surface cooling over China. However, the surface albedo feedback change has large seasonal and spatial variations. The decreased snow cover over the upper reaches of the Yellow River and the middle and upper reaches of the Yangtze River leads to an abnormal winter warming in MIROC-ESM (Lines 290–300).

(2) The authors find that there is decrease in clouds over China in the G4 experiment (Figures 7, S1 and S2). This decrease allows more net solar radiation at the surface. Some insight into the reason for the decrease in cloudiness in G4 should be provided in the revision.

**Response**: In summer, the SAI-induced decrease in cloudiness mainly occurs over northeastern and southeastern China. In winter, this decrease is coherent over China. The results indicate that the decreased cloud cover is related to the decreased latent heat flux under SAI forcing (Figs. 11c, f, and S1c). However, in summer, the effect of latent heat flux is partly offset by the SAI-induced moisture convergence at the troposphere in most models. The resultant increased cloud cover enhances the surface cooling over northwestern and central China (Fig. 11h). The relevant discussion has been added in the revision (Lines 284–290).

(3) Equations 8-11: How are these equations implemented in this work to estimate the 4 contributions discussed in section 4.3? This should be briefly discussed right after the derivation of these 4 equations. Also, the connection between these 4 equations and discussion in section 4.3 should be discussed in the beginning of section 4.3

**Response**: Considering the change in research method, this part has been shortened in the revision. Under SAI forcing, both the changes in atmospheric reflection and atmospheric absorption affect the downward clear-sky SW at the surface. We assume that the clear-sky atmospheric reflection change is only affected by atmospheric water vapor amount, and the clear-sky atmospheric absorption change is only affected by the aerosol scattering effect. Therefore, the change in downward surface clear-sky SW can be separated into two parts: the effects of solar radiation scattering ($SW_{SRM}$) and atmospheric water vapor amount ($SW_{WV}$). The relevant explanation has been added into Sect. 2.3 and the beginning of Sect. 4.3 (Lines 153–166, 269–271 and 278–282).

(4) Why does this focus on China? Why not the entire global domain? The rationale for the choice of the domain should be discussed in the revision.

**Response**: We agree that the global temperature response to SAI is of interest. We would like to mention that global-scale studies have been investigated systematically (e.g., Niemeier et al., 2013; Kashimura et al., 2017; Ji et al., 2018). Recently, increasing attention has been given to the climatic response to solar radiation modification on a regional scale (e.g., Pinto et al., 2019; Da-Allada et al., 2020; Xu et al., 2020). As the largest developing country in the world, China plays an important role in combating climate change. China's attitude to geoengineering is crucial to the international geoengineering research community. Considering the combined effect of the Tibetan Plateau and the East Asian monsoon, the climate over China would be strongly influenced by SAI. But the climatic impact has not yet been examined explicitly. For these reasons, we focus on China rather than the entire global in this study. The relevant explanation has been added accordingly (Lines 75–79).

In the revision, we have also changed our research method based on the comments of other reviewers. The diagnosis based on surface energy budget quantifies the effect of downward energy fluxes changes on surface air temperature over China under SAI forcing (Lines 136–153). The physical processes which dominate the temperature change have also been evaluated (Lines 265–300). We believe that this study provides new scientific insights.

**Specific comments:**

(1) Line 45: delete "simulating"

**Response**: Text revised (Line 51).

(2) Line 50: change "the decreasing" to "decrease".

**Response**: Text revised (Line 58).

(3) Lines 53-55: the overcooling of the tropics and undercooling of the polar regions would happen only if SRM is designed to offset the entire global mean surface temperature change. This important point should be included in the discussion here.

**Response**: This part has been deleted when we revise the manuscript.

(4) Lines 55-59: SRM does not address the ocean acidification problem caused by increasing levels of CO2 in the atmosphere. This deficiency of SRM should be also mentioned here.

**Response**: This deficiency has been added as you suggested (Lines 62–63).

(5) Line 70: Why do the author assess only surface temperature change? Why not the other important climate variables such as precipitation?

**Response**: SAI geoengineering aims to counteract anthropogenic global warming. Surface air temperature is thus a first-order variable that should be focused on. Therefore, this manuscript evaluates the impact of SAI on the surface air temperature over China and the underlying physical processes. As you mentioned, the effects of SAI on other climate variables, such as precipitation and monsoon, are also important. But they are not the objectives in this study.

(6) Line 79: change "the simulations in the G4 experiment" to "the G4 experiment".

**Response**: Text revised (==Line 96==).

(7) Line 79: provide a reference for the first phase of GeoMIP.

**Response**: Added accordingly (==Line 96==).

(8) Lines 108-118: What is the rationale for using the 1-layer atmosphere model in this study? What are its advantages and disadvantages? This should be briefly discussed.

**Response**: The single-layer model can help us separate the effects of atmospheric reflection fraction change and atmospheric absorption fraction change on SW. The former corresponds to the solar radiation scattering, and the latter corresponds to the atmospheric water vapor amount. This method has an inherent error due to the non-linear nature of equations. But this error is small and acceptable. This method is effective when analyzing the surface SW change under SAI forcing (Kashimura et al., 2017). The brief discussion has been added accordingly (==Lines 163–166==).

(9) Line 122: For clarity, change "R is the fraction of reflection" to "R is the fraction of solar radiation reflected by the atmosphere"

**Response**: Text revised (==Lines 161–162==). The $R$ has been changed into $F$ to avoid confusion.

(10) Line 155: "All the SCC are significant at the 99% level" How is this assessment made? Taylor diagram does not provide an assessment of the significance level of the correlation efficient. The method used for the statistical assessment should be briefly discussed.

**Response**: The significance of the spatial correlation coefficients is determined from the two-tailed Student's $t$-test. This expression has been added to the caption of Fig. 2 (==Lines 539–540==).

(11) Line 180: the lack of differences in trends between G4 and RCP4.5 is expected because the magnitude of the radiative forcing is the same in the experiment except in the

beginning when aerosols are suddenly injected in the G4 experiment.

**Response**: We agree with your point. That sentence has been revised accordingly (Lines 206–208).

(12) Line 205-208: The sign convention of LW is not clear in this paper. Is upward or downward LW is considered positive? This should be clarified in the revision.

**Response**: All the fluxes are defined as downward positive in the original manuscript. In the revision, we only consider the changes in downward surface radiative fluxes according to the surface energy budget equation. Those unclear expressions have been deleted accordingly.

(13) Lines 211-212: The figures show a decrease in clouds, but the text says clouds increase. The authors should carefully check their analysis.

**Response**: In summer, the regionally averaged changes in the cloud cover fraction over China show a consistent increase, although the decreased cloud cover occurs in some regions. In the revision, we have mainly discussed the robust change for individual regions (e.g., Lines 243–248). The regionally averaged change has been removed.

(14) Lines 215-216: The link between the deficit of downward LH and flux and increase in cloud cover is not clear. Either delete this discussion or provide clarity.

**Response**: This discussion has been deleted accordingly. In the revision, we consider that the increase in summer cloud cover is mainly related to the SAI-induced moisture convergence at the troposphere over northwestern and central China (Lines 287–290).

(15) Line 220-223: The discussion is unclear. Revise the text.

**Response**: In the revision, we only consider downward surface LW changes according to the surface energy budget equation. This part has been deleted accordingly.

(16) Lines 251-253: The message from this sentence is not clear. Revise the text for clarity.

**Response**: This sentence has been rewritten for clarity (Lines 274–276).

(17) Line 303: The Tilmes et al. 2018 paper discusses injection at multiple locations and not regional injections.

**Response**: The "regional injection" has been changed into "injection at multiple locations" accordingly (Lines 331–332).

(18) Figure 1: I believe cs and cl are interchanged in the illustration. Should be corrected.

**Response**: As you suggested, the "cs" and "cl" are interchanged in the original Fig. 1. In the revision, this illustration has been removed due to the change in research method.

(19) Figure 2: The last line of the caption: The oblique dotted line cannot be seen in the figure. Revise the figure or the caption.

**Response**: The statement of "the oblique dotted line" has been changed into "the dotted straight line" in the revision (Line 539).

**Reference:**

Da-Allada, C. Y., Baloïtcha, E., Alamou, E. A., Awo, F. M., Bonou, F., Pomalegni, Y., Biao, E. I., Obada, E., Zandagba, S., Tilmes, S., and Irvine, P. J.: Changes in west African summer monsoon precipitation under stratospheric aerosol geoengineering, Earths Future, 8, e2020EF001595, https://doi.org/10.1029/2020EF001595, 2020.

Ji, D., Fang, S., Curry, C., Kashimura, H., Watanabe, S., Cole, J. N., Lenton, A., Muri, H., Kravitz, B., and Moore, J.: Extreme temperature and precipitation response to solar dimming and stratospheric aerosol geoengineering, Atmos. Chem. Phys., 18, 10133–10156, https://doi.org/10.5194/acp-18-10133-2018, 2018.

Kashimura, H., Abe, M., Watanabe, S., Sekiya, T., Ji, D., Moore, J. C., Cole, J. N., and Kravitz, B.: Shortwave radiative forcing, rapid adjustment, and feedback to the surface by sulfate geoengineering: Analysis of the Geoengineering Model Intercomparison Project G4 scenario,

Atmos. Chem. Phys., 17, 3339–3356, https://doi.org/10.5194/acp-17-3339-2017, 2017.

Niemeier, U., Schmidt, H., Alterskjaer, K., and Kristjánsson, J. E.: Solar irradiance reduction via climate engineering: Impact of different techniques on the energy balance and the hydrological cycle, J. Geophys. Res.-Atmos., 118, 11905–11917, https://doi.org/10.1002/2013JD020445, 2013.

Pinto, I., Jack, C., Lennard, C., Tilmes, S., and Odoulami, R. C.: Africa's climate response to solar radiation management with stratospheric aerosol, Geophys. Res. Lett., 47, e2019GL086047. https://doi.org/10.1029/2019GL086047, 2020.

Xu, Y., Lin, L., Tilmes, S., Dagon, K., Xia, L., Diao, C., Cheng, W., Wang, Z., Simpson, I., and Burnell, L.: Climate engineering to mitigate the projected 21st-century terrestrial drying of the Americas: a direct comparison of carbon capture and sulfur injection, Earth Syst. Dynam., 11, 673–695, https://doi.org/10.5194/esd-11-673-2020, 2020.

---

## Author Response (AR2)

**Response to Report #2:**

**Overview:**

This study uses the GeoMIP G4 simulations to look at surface temperatures over China. The radiative effects are separated and analyzed singularly, leading to some interesting insight in the partitioning of the causes of the observed changes. The authors also carefully evaluate the models over the area of analysis for the historical period, which is always very commendable. I read the previous reviews, and I appreciate the changes performed by the authors (even if I don't really agree with the original reviews. Yes, exactly like there are studies focused on every region for climate change, any study focusing - and using local expertise - on regional impacts of geoengineering is important in my view). I think the manuscript is overall well written and robust. I have some minor comments below, but believe the manuscript is publishable for ACP.

**Minor Comments:**

(1) Line 25: In the text you say that the observed warnings are "weak and insignificant" (line 223). That's not what one would gather from this phrase in the abstract.

**Response**: The "warming" has been changed into "insignificant warming" for clarity (Line 24).

(2) Lines 67-68: The introduction is very clear and well written, but just a minor point: far more recent studies have shown that most of the ozone changes at low latitudes in the stratosphere are driven by dynamical changes in stratospheric circulation rather than heterogeneous chemistry, whereas SAD-induced changes are more dominant at high latitudes. See for instance Tilmes et al. (2022) and Visioni et al. (2021). Eastham et al. (2018) used solar reduction, hence it's not a good example here (see the comparison in Visioni et al. (2021) about solar dimming experiments versus the presence of stratospheric aerosols).

**Response**: As you suggested, the SAI-induced stratospheric ozone depletion is affected by both the heterogeneous chemistry and stratospheric circulation changes. We have rewritten this sentence and added appropriate references in the revision (Lines 65–67).

(3) Line 69: Maybe substitute "The" with "An appropriate SRM geoengineering strategy" just to make clear there can be multiple ones.

**Response**: Text revised (Line 68).

(4) Line 120: "perform" instead of "finish".

**Response**: Text revised (Line 119).

(5) Lines 327-331: The beginning of this paragraph is just wrong (lines 327-328) and needs some work. Equatorial injections tend to overcool the tropics because most of the aerosols are confined to the tropical stratosphere due to the strong confinement of the BDC. Eventually the aerosols are moved out of the tropical pipe - and allowed to reach higher latitudes. But what Kravitz et al. (2019) showed is that injecting at other latitudes in order to achieve a more comprehensive strategy where tropical overcooling is avoided tends to minimize many of the adverse impacts projected under equatorial injections. So the end of the paragraph is correct and well written, but it needs to be tied in with better wording over the stratospheric circulation.

**Response**: We have rephrased this part and added an appropriate reference in the revision as you suggested (Lines 326–329). Accordingly, the original reference (Kravitz et al., 2019) has been moved to Line 333.